# Examination of Lower Limb Microcirculation in Diabetic Patients with and without Intermittent Claudication

**DOI:** 10.3390/biomedicines11082181

**Published:** 2023-08-03

**Authors:** Katalin Biró, Barbara Sándor, Kinga Tótsimon, Katalin Koltai, Krisztina Fendrik, Dóra Endrei, Judit Vékási, Kálmán Tóth, Gábor Késmárky

**Affiliations:** 1First Department of Medicine, School of Medicine, University of Pecs, Ifjusag ut 13, H-7624 Pecs, Hungary; sandor.barbara@pte.hu (B.S.); totsimon.kinga@pte.hu (K.T.); koltai.katalin@pte.hu (K.K.); fendrik.krisztina@pte.hu (K.F.); endrei.dora@pte.hu (D.E.); toth.kalman@pte.hu (K.T.); kesmarky.gabor@pte.hu (G.K.); 2Department of Ophthalmology, School of Medicine, University of Pecs, Akác u. 1, H-7624 Pecs, Hungary; vekasi.judit@pte.hu

**Keywords:** diabetes mellitus, lower extremity artery disease, red blood cell aggregation, transcutaneous partial tissue oxygen pressure

## Abstract

Intermittent claudication is a frequent complaint in lower extremity artery disease, but approximately two thirds of patients are asymptomatic, most of which are diabetic patients. Non-invasive angiological and microrheological tests on diabetic subjects with and without intermittent claudication were performed in the present study. In total, 98 diabetic patients were included and divided into two groups: 20 patients (63.5 ± 8.8 years, 55% men, 45% women) had intermittent claudication, 78 patients (65.5 ± 9.3 years, 61.5% men, 38.5% women) were asymptomatic. Hand-held Doppler ultrasound examination, transcutaneous tissue partial oxygen pressure (tcpO_2_) measurement, Rydel–Seiffer tuning fork tests, and 6-min walk tests were performed, and erythrocyte aggregation was investigated. Ankle–brachial index (*p* < 0.02) and tcpO_2_, measured during provocation tests (*p* < 0.003) and the 6-min walk test (*p* < 0.0001), significantly deteriorated in the symptomatic group. A higher erythrocyte aggregation index and faster aggregate formation was observed in claudication patients (*p* < 0.02). Despite the statistically better results of the asymptomatic group, 13% of these patients had severe limb ischemia based on the results of tcpO_2_ measurement. Claudication can be associated with worse hemodynamic and hemorheological conditions in diabetic patients; however, severe ischemia can also develop in asymptomatic subjects. Non-invasive vascular tests can detect ischemia, which highlights the importance of early instrumental screening of the lower limbs.

## 1. Introduction

Based on the data of the International Diabetes Federation, 425 million people suffered from diabetes mellitus in 2017, and this number is increasing rapidly [1,2]. Diabetes can cause multiple complications, both at the macro- and microvascular levels. Even in newly discovered cases of diabetes, early diagnosis and detection of existing complications (e.g., retinopathy, nephropathy, neuropathy, peripheral artery disease) is important [3]. Microvascular complications may draw attention to the possible involvement of other vascular areas (e.g., coronary vessels, carotid or lower limb arteries) [4]. Lower extremity artery disease (LEAD) is a condition that develops when lower limb arteries become stenosed or occluded by atherosclerotic plaque(s). With a reduction in the arterial lumen, blood flow and oxygen supply to the muscles and other tissues decreases, resulting in pain, poor wound healing, and, in severe cases, tissue damage or gangrene. Based on walking impairment, pain, and ischemic ulcers, LEAD patients are traditionally classified by the Fontaine classification. Stage I includes patients who are asymptomatic, but for whom instrumental examination may reveal arterial steno-occlusive disease. Stage II refers to patients with claudication. In Stage IIA, intermittent claudication presents after more than 200 m of walking. In stage IIB, the walking distance is less than 200 m. Stage III describes rest pain. Stage IV includes patients with ischemic ulcers or gangrene.

LEAD is 2–4 times more frequent in diabetic patients than in the general population [5,6]. The 5-year mortality of LEAD patients is very high (30%), mainly due to major adverse cardiovascular events (e.g., atherosclerosis of the coronary or cerebral vessels) [7]. It is known that the survival of LEAD patients, even in the asymptomatic stage, is worse compared to non-vascular patients [8]. In addition, 20–30% of LEAD patients also have diabetes. The severity and duration of diabetes affect the development of the disease and its progression, and it has also been proven that there is a much greater opportunity of critical limb ischemia among patients with diabetes, and the major amputation rate of the lower limb is 7–15 times higher than in the non-diabetic population [9]. Diabetes is one of the main causes of lower limb amputation due to non-traumatic reason; moreover, patients may suffer another major or minor amputation within 3 years of the first amputation [10]. In patients with T2DM, involvement of the infrapopliteal arteries is more common with advanced calcification, multisegmental appearance, and poor distal outflow tract. LEAD can appear 10 years earlier in diabetic patients compared to the non-diabetic population. In later phases, critical limb ischemia (CLI) may develop with a high risk of major adverse limb and/or cardiac events (MALE, MACE) [11]. The diagnosis of CLI is confirmed by non-invasive vascular procedures: ABI < 0.4, ankle pressure < 50 mmHg, toe pressure < 30 mmHg or tcPO_2_ < 30 mmHg. International guidelines recommend WIFI classification, which gives a risk stratification for amputation based on the severity of the wound, ischemia, and foot infection [12,13]. Due to the possibility of asymptomatic (or silent) lower extremity artery disease in diabetes, it is important to perform vascular screening. Following arterial pulse palpation, the ankle–brachial index is the first non-invasive diagnostic tool with I/A evidence based on several guidelines [14,15]. The sensitivity and specificity of this test are high in non-diabetic subjects; however, due to media sclerosis in diabetes, renal failure, or severe calcification in the elderly, arteries become incompressible, resulting in false negative test results [16]. In this case, toe blood pressure or transcutaneous tissue oxygen pressure measurements could be good alternatives [17]. In addition to the vascular examination, the calibrated tuning fork test should be essential in the diabetic population, and can be easily performed as a screening method. Sensory neuropathy leads to the loss of different types of sensitivity: in addition to vibration, the sensation of pain is also impaired, which can mask tissue ischemia, and the classic claudication symptom may not present while walking; therefore, LEAD remains undiscovered [18]. In addition to macrovascular damage, microvascular involvement also plays a significant role, e.g., causing retinopathy, nephropathy. In addition to hemodynamic and endothelial factors, hemorheological changes also play a role in diabetic microvascular damage. Several studies have proven that hemorheological characteristics, such as erythrocyte deformability and aggregation, are significantly worse in the diabetic population compared to the non-diabetic population [19,20].

## 2. Materials and Methods

### 2.1. Focus and Aim of the Study

The aim of our study was to find out whether there is a significant difference in the angiological and the hemorheological results between diabetic patients with and without intermittent claudication. We investigated whether the ankle–brachial index can adequately detect lower limb ischemia in a diabetic population, and whether an additional non-invasive angiological method could identify more patients with peripheral arterial disease.

### 2.2. Participants

A total of 98 diabetic patients with previously known diabetic retinopathy were included in our study. These patients were followed regularly for their retinopathy at the Department of Ophthalmology. The participants were divided into two groups: 20 patients (63.5 ± 8.8 years, 50% men, 50% women) suffered from intermittent claudication, 78 patients (65.5 ± 9.3 years, 61.5% men, 38.5% women) were asymptomatic. Comorbidities, risk factors, concomitant medication, and physical status (including palpation of peripheral arteries, e.g., dorsal pedal artery (DPA), posterior tibial artery (PTA), popliteal, femoral artery) were recorded. Additional to the routine examination, hand-held Doppler examination, transcutaneous tissue partial oxygen pressure measurement, tuning fork tests, and 6-min walk tests were performed, and red blood cell aggregation was measured.

### 2.3. Non-Invasive Arterial Diagnostic Procedures

#### 2.3.1. Hand-Held Doppler Measurement, Ankle–Brachial Index Calculation

Hand-held Doppler ultrasound (MultiDoppy, 8 MHz, Medicad Ltd., HUN, serial number: 141203) and a manual sphygmomanometer were used to measure the systolic ankle pressure in both legs and arms; posterior tibial, dorsal pedal, and brachial arteries were checked. The cuff was placed around the ankle and on the upper arm on both sides. To calculate the ankle–brachial index (ABI), the higher systolic blood pressure was used as the denominator, while the higher pressure from the posterior tibial and dorsal pedal arteries at each ankle was considered as the numerator [13].

#### 2.3.2. Transcutaneous Tissue Oxygen Pressure

Transcutaneous partial tissue oxygen pressure (tcpO_2_) gives information regarding the partial oxygen pressure of the skin surface. Transcutaneous partial tissue oxygen pressure was measured after 15 min acclimatization at room temperature of the subject, while lying down, using a two-channel oximeter (Tina TCM 4000, Radiometer, Copenhagen, DEN). The measured site of the skin was cleaned with alcohol and shaved, when it was needed. A self-adhesive fixation ring was placed on the skin and filled with contact fluid. For the reference measurement, the first electrode was placed on the right side of the chest in the subclavicular region; then, the electrode was repositioned to the lateral part of the leg, and another electrode was fixed on the dorsal part of the foot at the first intrametatarsal space. The measurement was carried out at 44 °C locally, a temperature that causes maximum vasodilation [21]. The measurement at rest was followed by a test performed while elevating and hanging the leg.

#### 2.3.3. Calibrated Tuning Fork Test

To check diabetic polyneuropathy, the Rydel–Seiffer calibrated 128 Hz tuning fork test was performed on the radius and big toe on both sides [22,23]. The patient was asked to report the timepoint at which vibration disappears.

#### 2.3.4. 6-Min Walk Test

A supervised 6-min walk test (6MWT) was completed by each patient. During the test, the patient’s maximum walking distance was recorded [24]. The 6MWT was performed indoors, in a straight corridor, and the walking path was 30 m in length.

### 2.4. Blood Sampling, Sample Preparation, Red Blood Cell Aggregation Measurement

Blood samples were processed within 2 h after vein puncture of the antecubital vein after 12 h fasting. Blood was collected in lithium heparin-coated Vacutainer tubes with a 21-gauge butterfly infusion set according to the hemorheological guidelines. For red blood cell (RBC) aggregation measurement, a LORCA (Laser-assisted Optical Rotational Cell Analyzer; R&R Mechatronics, Hoorn, The Netherlands) aggregometer was used. Oxygenized blood samples were used for the measurements, and the temperature was kept at 37 °C [25]. In brief, 1 mL of oxygenated blood was injected into a gap between the static inner cylinder (“bob”) and the rotating outer cylinder (“cup”), creating a simple shear flow. Red blood cell aggregation index (AI), RBC disaggregation threshold shear rate (γ), and t_1⁄2_ characterizing the time that elapses to reach half the maximal aggregation amplitude were measured [26].

### 2.5. Statistical Analysis

IBM^®^ SPSS^®^ Statistics version 23.0 and one-way repeated ANOVA statistical tests and Tamhane post hoc tests were used to evaluate differences between the groups after using the Kolmogorov–Smirnov test to check the normality of the data distribution. Multinominal linear regression and stepwise analyses of the data were performed to predict the presence of claudication from RBC aggregation data (AI, t_1/2_), considering the principle of multicollinearity. Data are shown as mean ± standard deviation (SD). Results are considered significant at *p* < 0.05.

## 3. Results

Patients with intermittent claudication had slightly, but not significantly, poorer glycemic control (*p* = 0.239), and smoking was more common than in asymptomatic patients. All patients suffered from type II diabetes. Mean duration of diabetes was 15 and 15.9 years in the groups. Patients underwent regular diabetes check-ups at our outpatient clinic twice a year. Patients receiving insulin treatment underwent regular check-ups every 3 months. The risk factors of peripheral vascular disease are the same as those of atherosclerosis. The main risk factors include male gender, old age, high cholesterol level, hypertension, smoking, and diabetes (Table 1). Minor risk factors can be low HDL cholesterol, high triglyceride level, significant obesity, sedentary lifestyle, stress, and positive family history. These risk factors are included in the anamnestic data, but were not indicated in the study. Other non-classical risk factors include abnormal hemorheological parameters, hyperuricemia, reduced GFR, and microalbuminuria. Apart from the usual daily activities (e.g., housework, shopping), none of the patients performed sports activities. Some diabetic patients received bedtime insulin treatment in addition to the recommended oral antidiabetic treatment, and some of them received intensified insulin treatment due to poor carbohydrate metabolism. In addition, they followed a diabetic diet with reduced carbohydrate content based on self-report. Characteristics of the study population are summarized in Table 1.

In 20% of patients with claudication, a very low ABI (<0.4) was found, which indicates severe limb ischemia; in 25% of patients, moderately deteriorated ABIs were measured (0.4–0.7). Their mean ABI was significantly lower compared to the asymptomatic patients (0.79 ± 0.38 vs. 1.01 ± 0.34, *p* < 0.02). Despite the statistically better results of the symptom-free patients, 16% of them had low absolute values, indicating severe or moderate limb ischemia. Every fifth patient had non-compressible peripheral arteries, which implies non-informative measurements regarding intraluminal pressure, potentially masking severe ischemia. Distribution of the ankle–brachial indices in various ABI ranges is shown in Table 2.

In the claudication group, severe ischemia was confirmed in 40% of the patients based on tcpO_2_ measurement (<30 mmHg), and in half of these patients, a critically low value was revealed (<10 mmHg). In the asymptomatic group, good tissue oxygenation (>50 mmHg) was measured only in 19 patients (24.6%), while 12 patients (15.4%) had severely low tissue oxygen pressure (<30 mmHg) (Table 3).

We examined the ankle–brachial indices in connection with the low tissue oxygen tension (<30 mmHg) values. In the group showing claudication, seven out of nine patients with severely (<0.4) or moderately decreased (<0.7) ABIs were identified with severe ischemia via the tcpO_2_ measurement. In the asymptomatic group of patients with severe (<0.4) or moderate (<0.7) ABI reduction, 3 out of 13 people were certified with severe ischemia.

In total, 60% of the asymptomatic patients with borderline or normal ABIs (0.9–1.4) had low tissue oxygen tension values; precisely, 7 patients out of 30 were diagnosed with severe ischemia. Two patients in both groups were found with non-compressible arteries (ABI > 1.4) and severe tissue ischemia confirmed by the tcpO_2_ measurement.

These results are summarized in Table 4.

The absolute tcpO_2_ values in resting position measured at the leg and the foot were more deteriorated in the symptomatic than in the asymptomatic group. Foot elevation as a provocation test could worsen the pre-existing ischemia (Table 5).

The tuning fork values were under the normal range (6–8) in both groups. Significantly reduced results were found for the hallux of the diabetic patients with claudication (Table 6).

Symptomatic patients had significantly lower maximal walking distances during the 6-min walk test (192.33 ± 120 m vs. 310.31 ± 97.07 m; *p* < 0.0001).

Red blood cell aggregation index was significantly higher in symptomatic patients: the aggregate formation was faster and disaggregation shear rate was higher in this group. The nominal regression analyses regarding the prognostic power of RBC aggregation parameters for the presence of claudication revealed that the variables AI and t_1/2_ predicted the dependent variable, F(1,1) = 85, with statistical significance, *p* < 0.002. The results of these micro-hemorheological parameters are summarized in Table 7.

## 4. Discussion

Diabetes mellitus is associated with a higher risk of peripheral artery disease, which can develop earlier, progresses more rapidly than in non-diabetic patients, and causes multisegmental appearance on both lower extremities. Early detection of LEAD among diabetic patients has a crucial importance because of the high incidence of life-threatening cardio-, cerebro-, and peripheral vascular complications. Due to incompressible arteries and peripheral neuropathy, LEAD can stay hidden, and diagnostic procedures may be misleading. Diabetic retinopathy as a microvascular complication of diabetes was proven to be an independent risk factor for LEAD [27,28].

In our study, we aimed to assess the prevalence of LEAD among diabetic patients with microvascular complications (i.e., diabetic retinopathy), and to reveal the differences between symptomatic and asymptomatic diabetic subjects, using routine and alternative non-invasive vascular tests complemented by red blood cell aggregation parameters that can alter microvascular circulation.

Since diabetes is nearly asymptomatic in the early stages, approximately 50% of diagnoses are late, resulting in unnoticed development and progression of complications [29]. In a large prospective study (UKPDS 59), the association between HbA1c and retinopathy was revealed. The researchers found that hyperglycemia represents a higher risk of LEAD in patients with retinopathy, which may be a marker for vascular dysfunction, and can draw attention to the disease even before the diagnosis of diabetes [30]. In our study, the prevalence of LEAD based on ABI was equally high in both patient groups. The role of peripheral neuropathy and LEAD in the development of diabetic foot and lower extremity ulcers has long been known: neuropathy can hide (mask) lower limb ischemia. Approximately 1/3 of diabetic patients develop diabetic neuropathy, creating the basis of diabetic foot syndrome and neuropathic complaints [31,32]. It is symmetrical and has a distal appearance, and the lower limbs are most frequently affected [33]. Polyneuropathy is responsible for more than half of all limb amputations, and it has high economic and quality-of-life costs. Therefore, screening is essential for diagnosis, patient education, optimized glycemic control, and improved foot care. Screening for neuropathy can be carried out easily using the calibrated tuning fork or monofilament test. In our clinical study, the calibrated tuning fork test showed abnormal values, indicating neuropathy in more than half of the examined patients. Our results are in accordance with the literature: half of diabetic patients develop neurological damage, often causing symptoms of distal symmetric polyneuropathy [34]. We found a significant difference in the results of the tuning fork test between the two groups. In general, symptomatic patients are in a more advanced stage of LEAD. Although neuropathy may reduce the patient’s sensation of ischemia, the vasomotor damage may enhance microcirculatory disturbances, leading to impaired tissue oxygenation and provoked ischemic pain upon exertion (e.g., walking).

LEAD may have histological implications as well, as suggested by a study describing the small vessels of diabetic individuals with known peripheral artery disease [35]. They found that microvascular involvement may magnify peripheral resistance and intensify arterial atherosclerosis. A low ABI (below 0.9), indicating LEAD, is an independent predictor of cardiovascular diseases, and is a risk factor for cardiovascular morbidity and mortality [36]. In a Hungarian nationwide study, hypertensive patients between 50 and 75 years of age were examined. Based on the ABI measurement, 14.4% were diagnosed with PAD [37]. In another clinical trial, prevalence of LEAD was even higher, and exceeded 20% when its definition was based on an abnormal ankle–brachial index (ABI) [6,38].

Among the multiple complications of diabetes, diabetic foot is of particular importance due to its severity and often unfavorable outcome [39]. LEAD is one of the major pathologies that account for diabetic foot: impaired circulation, superinfections, and neuropathy can lead to the development of non-healing ulcers and amputation. LEAD was present in 49% of patients with diabetic foot in the EURODIALE study, and one-third of the study population had both LEAD and infection [40]. In the ADVANCE study, the prevalence of LEAD was determined based on a very strict protocol (i.e., a surgical and endovascular peripheral revascularization procedure or at least one amputation of the lower limb or chronic leg ulcer due to arterial stenosis), and was estimated to be 4.6% [41]. In the Hungarian nationwide large retrospective study (HUNVASCDATA project), Kolossvary et al. found that the lifetime risk of ulcer formation in the diabetic population is 25%. The amputation was preceded by the development of an ulcer in 85% of all cases, and 50.4% of the patients were diabetic [9]. In our study, the proportion of patients living with ulcers was similar: 10% of patients with claudication and 11.5% of the asymptomatic population. A larger portion of the claudication group had prior amputation compared to patients without claudication.

The prevalence of LEAD increases with the duration of diabetes, as shown by the UK Prospective Diabetes Study (UKPDS). At the time of diagnosis, only 1.2% of patients had LEAD, while 18 years later, the prevalence increased up to 12.5% [30]. The study by Marso et al. described that one in three people with diabetes over the age of 50 suffer from LEAD [42]. In our study, the mean duration of T2DM was 15–15.9 years, and LEAD was already present in 30% of patients with claudication and 15.4% of the asymptomatic population. The high incidence of LEAD in our study can be explained by the fact that our patients already had microvascular complications.

Although the gold standard diagnosis of LEAD is invasive angiography, guidelines recommend hand-held Doppler ultrasound examination as a first non-invasive vascular diagnostic method. The technique is a commonly accepted tool in microcirculation investigation and in ABI measurement of patients with presumed LEAD [13,43]. Although it has great value in determining non-diabetic LEAD, it can produce unreliable results in diabetes cases. In a review, Xu et al. found that the sensitivity of the Doppler method ranges between 15 and 79% when detecting moderate or severe lower extremity stenosis. On the other hand, the sensitivity was lower in diabetic and elderly patients. The specificity of the Doppler technique is very high, ranging from 83.3 to 99% in various studies [44]. This screening test is fundamental, and has a high importance from an epidemiological point of view, since the survival of LEAD patients is significantly worse compared to non-vascular patients [8]. The test can indicate peripheral artery disease even in an asymptomatic state, usually with an ABI value between 0.75–0.9. If ABI is abnormal, as with advanced atherosclerosis, usually 50–70% lower limb arterial stenosis can be verified by invasive angiography [14].

Based on the AHA/ACC 2016 recommendation, in patients with normal (1.00–1.40) or borderline (0.91–0.99) ABI in the setting of non-healing wounds or gangrene, it is reasonable to diagnose CLI by using the toe–brachial index (TBI) with waveforms, tcpO_2_, or skin perfusion pressure (SPP) measurement. In PAD with an abnormal ABI (≤0.90) or with non-compressible arteries (ABI > 1.40 and TBI ≤ 0.70) in the setting of non-healing wounds or gangrene, TBI with waveforms, tcpO_2_, or SPP can be useful to evaluate local perfusion [45]. Measurement of tcpO_2_ is a non-invasive method for detecting microcirculation and tissue oxygen supply. It can give information about skin perfusion and ischemia, which can be used to define the degree of lower extremity arterial disease [17]. Our results from the tcpO_2_ measurements show significantly lower tcpO_2_ values measured for the leg and the foot at rest and at elevation in symptomatic patients compared to the asymptomatic group. This result supports the assumption of the more severe macro- and microcirculatory disorders in diabetic patients with claudication: cramps as a sign of ischemia can be objectified with tcpO_2_ measurement. Our study shows that the hand-held Doppler technique plus the tcpO_2_ measurements clearly identify PAD patients in both groups: in the borderline (0.9–1) and normal (1–1.4) ABI ranges, a high portion of asymptomatic patients (41.67% and 16.67%, respectively) had severe limb ischemia (tcpO_2_ < 30 mmHg). A portion of patients with non-compressible arteries (ABI > 1.4) also had severe ischemia: 25% in the claudication and 16.67% in the asymptomatic group. With the combination of the two methods, we identified 13 new patients with previously hidden severe limb ischemia, which accounts for 13% of the entire study population.

In addition to the severe hemodynamic and neuropathic abnormalities, hemorheological parameters also play a significant role in diabetic patients due to the poor distal outflow. In the study by Dupuy-Fons et al., significantly deteriorated hemorheological parameters (blood viscosity, RBC aggregation, RBC rigidity index, hematocrit/viscosity ratio, fibrinogen level) were presented in diabetic patients that required major amputation (above or below knee) compared to other diabetic patients who did not need major amputation (no amputation or only toes amputation) [46]. In a previous study, our team confirmed that patients suffering from diabetic retinopathy had more increased RBC aggregation compared to both an age-matched non-diabetic population and a young, healthy control group [19]. In the present study, increased red blood cell aggregation and faster aggregate formation were proven in patients with claudication compared to asymptomatic patients. Based on our findings, RBC aggregation parameters can predict the presence of claudication as well. These results may confirm that hemorheological factors have a strong influence on blood flow in LEAD patients. Ernst and coworkers described the relevance of rheological factors in the maintenance of pain-free walking distance in LEAD patients. Researchers found that physical activity had a beneficial effect on hemorheological parameters, leading to increased nutritive flow in the hypoxic vascular bed [47]. Our results support these previous findings: the higher RBC aggregation is, the more probable that claudication will occur. This could result from the diminished oxygen delivery to distal tissues in the case of hemodynamic and hemorheological disturbances, and may have prognostic significance in chronic peripheral occlusive arterial disease in advanced cases.

## 5. Conclusions

There were fewer patients in the claudication group; the sensitivity of the study could have been increased if an equal number of patients had been included in both groups. Toe pressure was not measured due to unavailability at the beginning of this study.

## 6. Summary

Ischemic symptoms may not be recognized, neither by patients with diabetic neuropathy, nor by the physicians performing ABI measurements, due to the false negative results. Thus, supplementary non-invasive diagnostic tools (e.g., transcutaneous partial oxygen pressure and toe pressure measurements) are needed as an alternative diagnostic step to clarify the patient’s condition [21]. Limb loss is a severe complication of LEAD, causing impaired quality of life. Therefore, early recognition of ischemia using non-invasive vascular tests (screening) could play a key role in the prevention of diabetic patients.

In the case of diabetes, a combination of Doppler and tissue oxygenation measurements may improve diagnostic power. Diabetic patients without claudication should also be screened for LEAD, since neuropathy could obscure the complaints and, together with a deteriorated rheological status, enhance disease progression. Our investigation revealed higher and faster red blood cell aggregation, more severe neuropathy, and lower tissue oxygenation in diabetic patients with claudication, all of which indicate seriously impaired macro- and microcirculation. These facts evoke attention to an early, multitool screening procedure in diabetic patients with and without claudication.

## Figures and Tables

**Table 1 biomedicines-11-02181-t001:** Characteristics of the study population.

	Symptomatic (%) (*n* = 20)	Asymptomatic (%) (*n* = 78)
Male	55	61.5
Female	45	38.5
Smoking habits	30	7.7
Mean duration of DM (years)	15	15.9
Mean HbA1c ± SD (%)	8.16 ± 1.69	7.44 ± 1.5
Chronic coronary syndrome (CCS)	35	29.4
Cerebrovascular disease (CVD)	5	7.7
CCS + CVD	10	--
PAD (previously diagnosed)	30	15.4
Dyslipidemia	75	57.7
Hypertension	100	78.2
Ulcer/gangrene	10	11.53
Minor/major amputation	20	3.85

**Table 2 biomedicines-11-02181-t002:** Distribution of ankle–brachial index in the study population.

ABI Range	Symptomatic Patients (%) *n* = 20	Asymptomatic Patients (%) *n* = 78
<0.4	20% (*n* = 4)	6.41% (*n* = 5)
0.4–0.7	25% (*n* = 5)	10.26% (*n* = 8)
0.7–0.9	10% (*n* = 2)	24.35% (*n* = 19)
0.9–1.0	25% (*n* = 5)	17.95% (*n* = 14)
1.0–1.4	15% (*n* = 3)	20.51% (*n* = 16)
>1.4	5% (*n* = 1)	20.51% (*n* = 16)

**Table 3 biomedicines-11-02181-t003:** Transcutaneous partial tissue oxygen pressure values of patients with or without intermittent claudication.

tcpO_2_ Values	Symptomatic Patients (%)	Asymptomatic Patients (%)
>50 mmHg	15	24.6
30–50 mmHg	45	60
10–30 mmHg	20	15.4
<10 mmHg	20	-

**Table 4 biomedicines-11-02181-t004:** Distribution of ankle–brachial indices with severe limb ischemia based on partial tissue oxygen pressure values (<30 mmHg) in the study population.

ABI Range	Symptomatic Patients (%) *n* = 8	Asymptomatic Patients (%) *n* = 12
<0.4	4 (50%)	1 (8.33%)
0.4–0.7	2 (25%)	2 (16.67%)
0.7–0.9	-	-
0.9–1.0	-	5 (41.67%)
1.0–1.4	-	2 (16.67%)
>1.4	2 (25%)	2 (16.67%)

**Table 5 biomedicines-11-02181-t005:** Results of tcpO_2_ measurement.

Position of the Electrode	Symptomatic Patients	Asymptomatic Patients	*p* Values
Chest	51.64 ± 19.09	52.47 ± 12.64	*p* = 0.831
Leg at rest	41.11 ± 17.05	49.50 ± 11.45	*p* = 0.018
Leg at elevation	38.08 ± 16.54	44.26 ± 11.46	*p* = 0.119
Leg at stasis	57.5 ± 14.64	58.58 ± 12.83	*p* = 0.795
Foot at rest	31.70 ± 17.47	43.33 ± 11.53	*p* = 0.001
Foot at elevation	26.00 ± 18.41	39.78 ± 14.56	*p* = 0.003
Foot at stasis	45.71 ± 19.99	53.00 ± 14.75	*p* = 0.122

**Table 6 biomedicines-11-02181-t006:** Results of the calibrated tuning fork test.

Localization	Symptomatic Patients	Asymptomatic Patients	*p* Values
Right hallux	3.36 ± 2.06	4.45 ± 1.92	*p* = 0.045
Left hallux	3.28 ± 2.05	4.54 ± 1.99	*p* = 0.038
Proc. styl. radii	5.85 ± 1.02	6.41 ± 1.31	*p* = 0.149

**Table 7 biomedicines-11-02181-t007:** Results of red blood cell aggregation.

Parameters of Aggregation	Symptomatic Patients	Asymptomatic Patients	*p* Values
AI	71.94 ± 5.93	65.78 ± 6.78	*p* < 0.001
t_1/2_ (s)	1.36 ± 0.54	1.94 ± 0.71	*p* = 0.002
γ	197.11 ± 80.29	138.12 ± 41.58	*p* < 0.001

## Data Availability

Not applicable.

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
