# Peer review of "Examination of Lower Limb Microcirculation in Diabetic Patients with and without Intermittent Claudication"

_biomedicines, 2023, doi:10.3390/biomedicines11082181_

Round 1

Reviewer 1 Report

Dear Authors

you report that Claudication could be associated with 24 worse hemodynamical and hemorheological status in diabetics; nevertheless, severe ischemia can develop even in asymptomatic subjects which can be revealed by noninvasive vascular tests, which highlights the importance of the early instrumental screening of the lower limbs.

These data are crucial for screening of the lower limbs but we would like to known if there are differences in lyfestile (exercise time) and diet, and salt intake and if these differences are reported by sex/gender differences

Many thanks 

Dear Authors

you report that Claudication could be associated with 24 worse hemodynamical and hemorheological status in diabetics; nevertheless, severe ischemia can develop even in asymptomatic subjects which can be revealed by noninvasive vascular tests, which highlights the importance of the early instrumental screening of the lower limbs.

These data are crucial for screening of the lower limbs but we would like to known if there are differences in lyfestile (exercise time) and diet, and salt intake and if these differences are reported by sex/gender differences

Many thanks 

Author Response

Answer to Reviewer 1 per report biomedicines-2445905

Dear Reviewer!

Thank you very much for your valuable work.

1) These data are crucial for screening of the lower limbs but we would like to know if there are differences in lifestyle (exercise time) and diet, and salt intake and if these differences are reported by sex/gender differences. Many thanks

All patients performed their usual daily activities (e.g., housework, shopping), none of them performed other regular physical training. Some diabetic patients received bedtime insulin treatment in addition to the recommended oral antidiabetic treatment, and some received intensified insulin treatment due to poor carbohydrate metabolism. In addition, they followed a diabetic diet with reduced carbohydrate content based on the anamnestic data. All patients suffered from hypertension, therefore salt restriction is a part of non-pharmacological treatment, but detailed information on salt intake was not available.

The required editing of English language has been performed.

Sincerely Yours,

Katalin Biro

Reviewer 2 Report

In this study, these authors report that diabetic patients with intermittent claudication had worse results than who did not complain regarding a serie of tests aiming to detect lower limb arterial disease.

I have a serie of remarks:

Please define LEAD.

The patients were selected on the basis of a retinal disease. It is the most appropriate way to select patients with microvascular disease, and therefore a bias to study prevalence of given functional anomalies for distal vascularization in patients with diabetes.

No characteristics of diabetes are given: Type 1 or 2 or others, diabetes duration and control, demographics, risk factors, etc...

Author Response

Answer to Reviewer 2 per report biomedicines-2445905

Dear Reviewer!

Thank you very much for the evaluation and your special attention to our article. I answer your questions as follows.

1) Please define LEAD.

Lower extremity artery disease (LEAD) is a condition that develops when lower limb arteries become stenotic or occluded by atherosclerotic plaque(s). With reduction of the arterial lumen, blood flow and oxygen supply to the muscles and other tissues decreases resulting in pain, poor wound healing, and in severe cases tissue damage or gangrene.

The definition has been included in the manuscript.

2) The patients were selected on the basis of a retinal disease. It is the most appropriate way to select patients with microvascular disease, and therefore a bias to study prevalence of given functional anomalies for distal vascularization in patients with diabetes.

Diabetes has many vascular complications causing both micro- and macrovascular damages. Retinopathy is a frequently recognized complication of diabetes due to the relatively easy access to ophthalmological screening, while other complications (e.g., peripheral angiopathies and polyneuropathy) may progress into an advanced stage without recognition in the lack of screening. Therefore, we focused on a diabetic population in this study who were regularly followed and at least one complication (retinopathy) had already been revealed. This could be associated with the higher prevalence of (frequently hidden) peripheral angiopathy; a need for systematic vascular screening is emphasized.

3) No characteristics of diabetes are given: Type 1 or 2 or others, diabetes duration and control, demographics, risk factors, etc…

Characteristics of diabetes (type, duration, risk factors, demography, frequency of check-ups) have been added to the manuscript.

Sincerely Yours,

Katalin Biro

Round 2

Reviewer 1 Report

Dear authors it is very crucial paper

but we don't known if there gender differences and age differences in two groups.

However can you indicate the exercise role in these data (3 a week, 5 a week etc?) and alchool intake ?

thank you 

Dear authors it is very crucial paper

but we don't known if there gender differences and age differences in two groups.

However can you indicate the exercise role in these data (3 a week, 5 a week etc?) and alchool intake ?

thank you 

Author Response

Answer to Reviewer 1 per report Biomedicines-2445905

Dear Reviewer!

Thank you very much for your valuable work, evaluation and suggestions for additions! You can see the answers to your questions below.

1. Dear authors it is very crucial paper but we don't known if there gender differences and age differences in two groups.

Answer: I presented the gender and age distribution of the two groups in the abstract and in the methodological chapter.

98 diabetic patients were included: 20 patients (63.5±8.8 yrs, min 54 yrs, max 76 yrs; 55% men, 45% women) had intermittent claudication, 78 patients (65.5±9.3 yrs, min 37 yrs, max 80 yrs;61.5% men, 38.5% women) were asymptomatic. There are no significant differences between the two groups regarding age (p = 0.672) or gender (p = 0.545).

2. However can you indicate the exercise role in these data (3 a week, 5 a week ec?) and alcohol intake?

Answer: The patients in the present study did not perform regular physical exercise either on an institutional supervised nor on a home-based manner. They only walk as part of their daily routine approximately 3 times a week: shopping, working around the house, maybe a short walk. Based on the anamnestic data, an average consumption of 1-2 units of alcohol per day can be expected, although there were patients who did not consume alcohol at all or only ocaasionally (e.g. holidays, family events) The method section also has been improved, supplemented with additional information. I supplemented one more reference.

The required editing of English language has been performed.

Sincerely Yours,

Katalin Biro

Reviewer 2 Report

No additional comment

Author Response

Answer to Reviewer 2 per report Biomedicines-2445905

Dear Reviewer!

Thank you very much for the evaluation, your kind advice, your addition and special attention to our article.

Sincerely Yours,

Katalin Biró

Round 3

Reviewer 1 Report

ok